# Heating, Ventilation, and Air Conditioning (HVAC) Noise Detection in Open-Plan Offices Using Recursive Partitioning

**Gino Iannace *, Giuseppe Ciaburro and Amelia Trematerra**

Dipartimento di Architettura e Disegno Industriale, Università degli Studi della Campania Luigi Vanvitelli, 81031 Aversa, Italy; giuseppe.ciaburro@unicampania.it (G.C.); amelia.trematerra@unicampania.it (A.T.)

\* Correspondence: gino.iannace@unicampania.it; Tel.: +39-0815010700

**Abstract:** Open-plan offices have lower construction costs, allowing for significant savings in space and, according to designers, facilitate communication between workers, thus, improving collaboration, as well as the exchange of ideas. For these reasons, this type of office has become widespread, while highlighting numerous limitations and various problems. These include the control of anthropic and electromechanical noise. In this study, measurements of the noise emitted by a heating, ventilation, and air conditioning (HVAC) system were carried out in an open-plan office. The average spectral levels in a 1/3 octave band were compared through correlation analysis, to identify any redundant data. A model was then adapted to evaluate the importance of the variables, in order to classify the characteristics, by importance. To reduce the number of predictor variables, a selection analysis of the characteristics was carried out. A subset of predictors was extracted to be used to produce an accurate prediction model. Finally, a model based on recursive partitioning, to detect the operating conditions of an HVAC system, was developed and applied, so as to provide insights into the development and application of this technique, in these contexts. The high accuracy of the model (Accuracy = 0.9981) suggests the adoption of this tool for several applications.

**Keywords:** open plan office; workplace noise; HVAC noise; recursive partitioning; feature selection; random forest

---

## 1. Introduction

Nowadays, the organization of the working hours has been heavily influenced by globalization. Due to the ever-increasing competition in the market, employers tend to demand productivity gains and increasingly heavy work rates from their employees. People, on the other hand, tend to identify themselves more and more with their work, employing all their energies in it. However, to be able to do the best in our work, it is necessary that the work environment is as comfortable as possible. Much psychic pathology (stress, panic, anxiety, etc.) can be derived from an inadequate working environment, generating discomfort in individuals and interfering, negatively, with their ability to express their potential.

The essential elements that must be taken into account when creating a comfortable work environment are, lighting, acoustic, and thermal comfort. Noise in the workplace has become one of the most important problems. The introduction of continuous technological processes has led to the multiplication of noise sources and an increase in the percentage of workers exposed to this risk factor. This problem not only concerns industrial environments where there are very noisy sources, but also applies to quiet working environments, such as offices. Due to the nature of the work carried out in such environments, a particularly quiet environment is needed, in order to allow concentration and communication.

An architectural solution that has been very successful over the last few years is represented by open-plan offices. They have lower construction costs, allowing for significant space savings and, according to designers, facilitate communication between workers, thus, improving collaboration, as well as the exchange of ideas. Unfortunately, many studies show that few workers love to work in them since the buzz of chatter is distracting and there is the lack of privacy that is necessary to work quietly [1–3].

Although office workers are exposed to noise levels well below the thresholds of action imposed by the legislation, it has been calculated that a worker decreases productivity by 66% when exposed to a colleague's conversation [4]. A recent study found that most workers prefer private offices to open-plan offices because open spaces have a negative influence on employee concentration, privacy, and emotional well-being [5]. Evans et al. found that simulated low-intensity, open-office noise, produced elevated urinary epinephrine levels in workers, lowered the post-noise-exposure task performance (indicative of a depressed motivation), and reduced the use of ergonomic work-furniture features designed to provide opportunities for postural adjustments during work [6]. Perrin Jegen et al. conducted a study that establishes a link between the overall satisfaction of employees with their work environment and the perception of their health.

The less employees are satisfied with their work environment, the less they are considered healthy. The annoyance deriving from the sound sources represents an inconvenience, but it is not sufficient to explain the overall average satisfaction of employees, with respect to their work environment. Lack of privacy and control over the work environment are equally perceived problems. A calmer environment, with the possibility of holding private conversations and being out of sight of others, represent the most significant requests [7]. A study on the occupant survey database from Center for the Built Environment (CBE), indicated that occupants assessed Indoor Environmental Quality (IEQ) issues in different ways, depending on the type of the workspace. Enclosed private offices outperformed open-plan layouts in acoustics, privacy, and proxemics issues.

Benefits of enhanced 'ease of interaction' were smaller than the penalties of increased noise level and decreased privacy, resulting from open-plan office configuration. The advantages deriving from the simplicity of interaction have been evaluated below the problems deriving from the increase in noise level and the reduction in privacy of an open-space office [8].

Haapakangas et al. have used distraction distance, the spatial decay rate of speech, speech level at 4 m from the speaker, and the average background noise level, as possible predictors of perceived noise disturbance. The results show that an increase in distraction distance predicts an increase in disturbance by noise. Other predictors may not be associated with noise disturbance, alone. The results confirm that a good acoustic design of such environments can improve the working conditions in such environments [9].

The sources of noise in the office are not only anthropogenic but are also electromechanical. The former sources are of a discontinuous nature, over time, while the electromechanical ones can be of a discontinuous type (telephone ringing, photocopy and telefax) or continuous (heating, ventilation, and air conditioning (HVAC), computer), as well as characterized by considerable differences in terms of spectral contribution. Heating, ventilation, and air conditioning (HVAC) is the technology of indoor and vehicular environmental comfort. Its goal is to provide thermal comfort and acceptable indoor air quality.

In offices, a fan coil is often used as the terminal of an HVAC system. In a fan coil, the fan retrieves the ambient air (to be heated or cooled) by an opening located at the top of the terminal. Once in, the air is first filtered and then pushed by the fan towards the heat exchange coil. Here, by forced convection, this exchanges heat with fluid. In the heating mode, the heat is drawn; in the cooling regime the heat is transferred. In the case of the cooled air, a dehumidification process is activated simultaneously [10]. Figure 1 shows the fan coil structural elements.

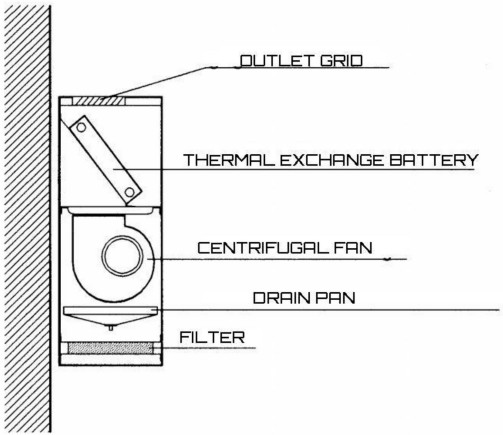

**Figure 1.** Fan coil structural elements.

In such systems, noise originates from the internal fan [11,12]. In very quiet places, such as offices, when the fan is activated after the thermostat has been activated, its switch-on causes the background noise level to pass from the quiet state to the active fan status [13,14]. In particular, in the condition of the active fan, there are two speeds available—low and high.

This paper presents the results of an experimental survey on the contributions to noise levels, produced in an open-plan office by an HVAC system. In the second part of the work, the measured data are analyzed, through different forecast models, to recognize the operating conditions of the HVAC system, automatically.

## 2. Methodology

The noise survey in the offices was carried out in an open-plan office located in a research center. The office had a size of, approximately, 100 m$^2$. In the office, there were four groups of stations, as shown in Figure 2.

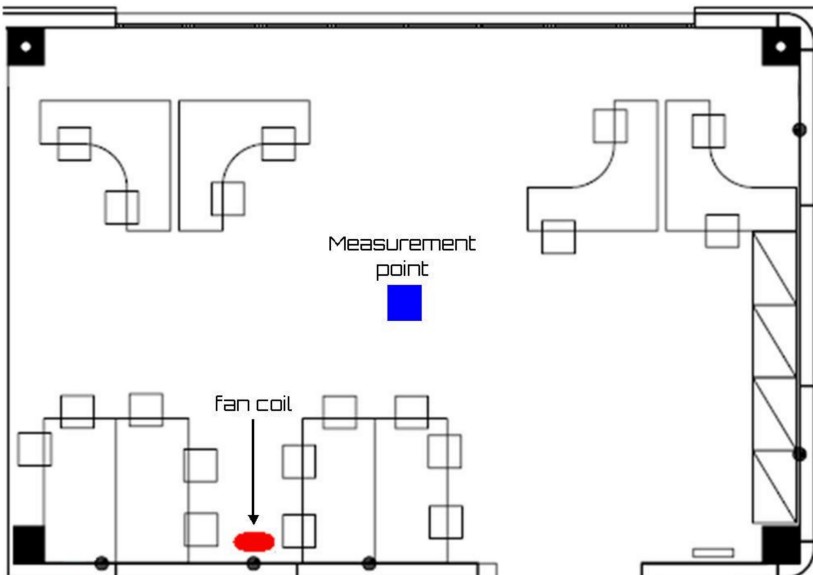

**Figure 2.** Open-plan office map. Four groups of stations are clearly distinguishable; the position of the HVAC and the measurement point is also indicated. The office was 7.4 m wide, 11.0 m long, and 4.0 m high. The furnishing accessories are almost all reflective, except for the curtains near the window, which are of absorbent material.

In each work place, preliminary noise measurements were carried out, in order to characterize the noise produced by the air conditioning terminals. The most exposed location was chosen as a measurement point. The noise level of the terminal was evaluated in the minimum and maximum speed conditions of the electric fans.

The measurements of the noise levels present in the work stations were performed according to the international standard ISO 3382-3 [15], which specify the method of measuring the acoustic properties of open-plan offices.

An integrating sound level meter of the type SOLO 01 dB Metravib was placed in the operator's position by placing the microphone at 1.20 m from the floor, 30 cm away from the work table, and with an inclination of 30° from the horizontal plane. The instrument used, complied with the requirements of the IEC61672-2002 standard [16]; the sound level meter was configured for the acquisition of a level of a linear sound pressure equivalent to the statistical levels (weighted "A"), with a fast time constant and a frequency spectra of 1/3 of an octave.

All measurements were taken while ensuring that the other equipment or machinery in the environment were switched off. Furthermore, in order to characterize the source, each measurement was performed in the absence of people, but with the normal daytime background noise. After identifying the most exposed location, at this point the measurements were taken with a duration of 1 h, for each operating condition. In this location, the background noise was then measured for the same duration.

A first exploratory analysis was carried out to analyze the temporal history and the spectra of the three measurements, in order to identify the characteristics that are able to discriminate between the different operating conditions. A visual analysis was conducted through a wrapper, for the density lattice plots, to visualize the data [17].

The correlation analysis between the available variables was then conducted with the aim of identifying those that are most correlated with each other. When two variables are correlated, when one changes, so does the other. The absence of a correlation, on the other hand, implies that there is no correlation between the variables, which could, therefore, both be explanatory of the phenomenon under analysis. The Pearson correlation index was calculated that expresses a possible linear relationship between the variables. It is given by the sum of the products of the standardized scores of the two variables, divided by the number of subjects (or observations). This coefficient can assume values ranging from $-1.00$ (there is a perfect negative correlation between the two variables) and $+1.00$ (there is a perfect positive correlation between the two variables). A correlation of 0 indicates that there is no relation between the two variables [18].

A model to evaluate the importance of variables, in order to rank features by importance, was then built. Generally, importance provides a score that indicates how useful or valuable each feature was, in the construction of the model. The more an attribute is used to make the model, the higher is its relative importance. This importance is calculated explicitly for each attribute in the dataset, allowing attributes to be ranked and compared to each other. To train the model, a Learning Vector Quantization (LVQ) method was used. This method represents a supervised version of the vector quantization used, when labeled input data are available. This learning technique uses the class information to slightly reposition the Voronoi vectors, so as to improve the quality of the classifier decision regions [19–21].

To reduce the number of predictor variables, a feature selection analysis was conducted. This is a special form of reduction of the dimensionality of a given dataset—it is the process of reducing the inputs for the elaboration and the analysis or the identification of the most significant characteristics, with respect to the others. The selection of characteristics is necessary to create a functional model, i.e., a reduction of cardinality, imposing a limit higher than the number of characteristics that must be considered during its creation. This is useful when the data contains redundant information. A recursive feature elimination (RFE) algorithm is used. This algorithm implements a backwards selection of predictors, based on the predictor importance ranking. The predictors are ranked, and the

less important ones are sequentially eliminated, prior to modeling. The goal is to find a subset of predictors that can be used to produce an accurate model [22–26].

Finally, a recursive partitioning model was implemented in order to detect the operating conditions of the HVAC system. This model creates a decision tree to classify categorical data, by splitting them into sub-systems, based on several dichotomous independent variables. The process is named recursive because each sub-system may, in turn, be split an indefinite number of times, until the splitting process terminates after a particular stopping criterion is reached [27,28].

## 3. Results

Three operating conditions were analyzed: OFF (HAVC off = background noise), ON-L (HAVC on low speed), and ON-H (HAVC on high speed). Three different measurement sessions were carried out, lasting about one hour each. The time history of the three operating conditions are shown in Figure 3.

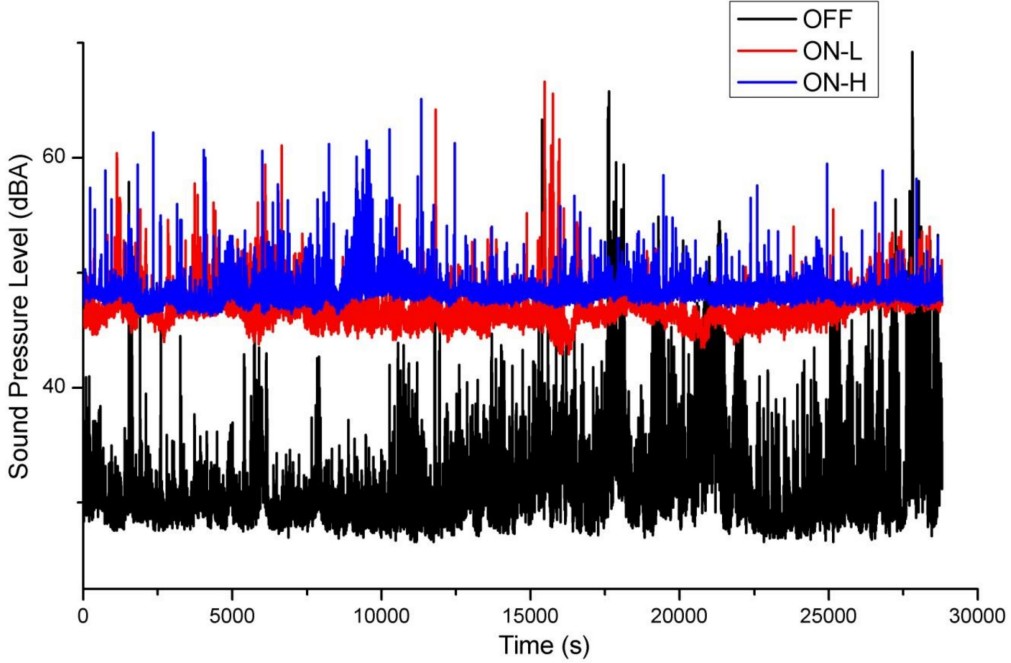

**Figure 3.** Time history of the three operating conditions (OFF (HAVC off = background noise), ON-L (HAVC on low speed), and ON-H (HAVC on high speed)).

Table 1 presents the Acoustic measurements for the operating conditions performed in the measurement session. From the analysis of the results shown in Figure 3 and Table 1, it is possible to note that the different operating conditions were confirmed by the noise levels recorded by the sound level meter. Under the operating conditions, with HVAC on (low and high speed), the noise levels were always higher (47.1 dBA–48.6 dBA), compared with the background noise (38.0 dBA–HVAC off).

**Table 1.** Acoustic measurements in an open-plan office.

| Operating Conditions | LeqA (dBA) | L95 (dBA) | L5 (dBA) | Standard Deviation |
|---|---|---|---|---|
| ON-H (HAVC on high speed) | 48.6 | 47.1 | 49.9 | 1.0 |
| ON-L (HAVCon low speed) | 47.1 | 44.9 | 49.0 | 1.3 |
| OFF (HAVC off = background noise) | 38.0 | 27.9 | 41.6 | 4.4 |

From the analysis of Table 1, it is also possible to note that under the operating conditions, with the HVAC system switched on, the values of the standard deviation were clearly lower than those recorded during the measurements of the background noise, meaning that in the first two cases the noise took on a predominantly stationary nature.

The results of the measurements were subsequently elaborated to look for the characteristics that were able to discriminate the different operating conditions of the HVAC system. In this first data analysis, it was decided to use the average spectral levels in 1/3 octave bands, between 20 Hz and 20 kHz, as the descriptors. Figure 4 shows the average spectral levels in 1/3 octave band between 20 Hz and 20 kHz, for each of the three observation periods measured.

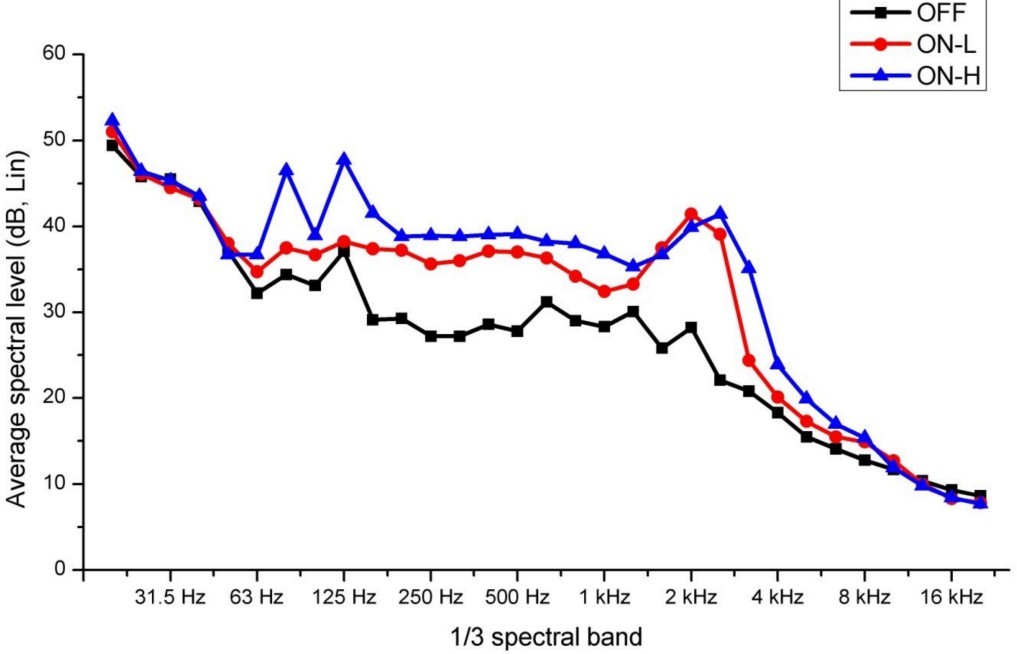

**Figure 4.** Average spectral levels in the 1/3 octave band, between 20 Hz and 20 kHz, for each of the three observation periods measured.

From the analysis of the results shown in Figure 4, it is possible to note that the operating conditions were easily identifiable, even through a simple visual analysis. Leaving aside the data related to the low and high frequencies, where the three curves overlap, but limiting the analysis to the interval that went from 50 Hz to 5 kHz, it was possible to note that the curves were clearly distinguished, allowing for the distinction between the different conditions of operation. It was possible to predict that the operating condition ON-H was the one with the highest spectrum values. Immediately below this curve, there were the values relating to the ON-L operating condition. Finally, below the two curves, clearly distinguished from them, was the curve related to the background noise.

To build a classification model of the operating conditions of the HVAC system, the measured data were processed by extracting 20 s observations. For each observation, the weighted equivalent levels A for each 1/3 octave band were calculated. Five hundred and forty observations were collected and thirty-one variables were selected. To identify the characteristics that were able to discriminate between the different operating conditions, a visual analysis was conducted through a wrapper, for the density lattice plots, to visualize the data (Figure 5).

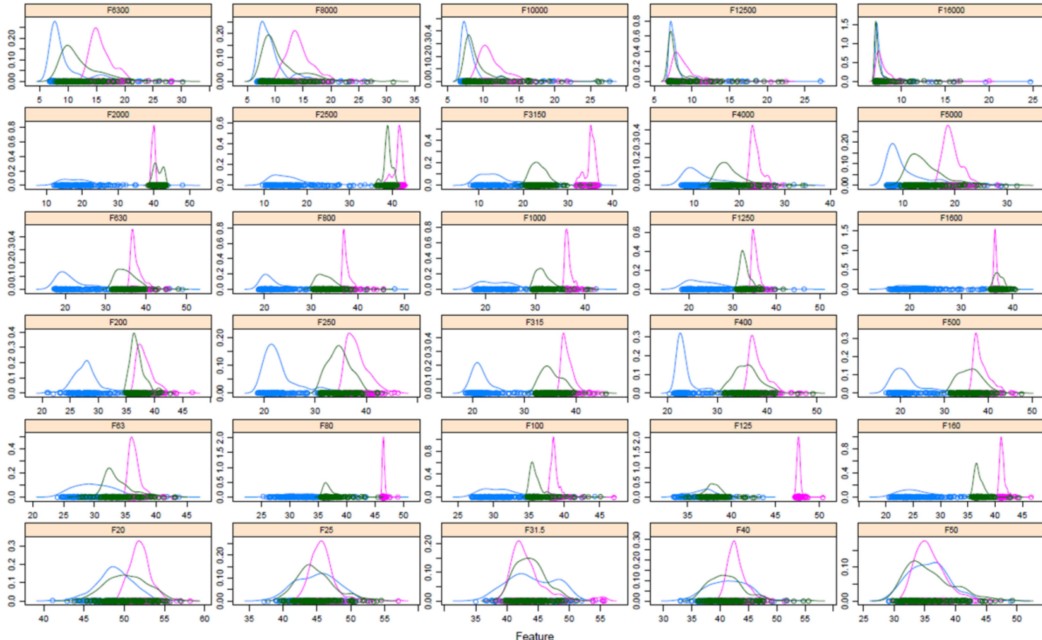

**Figure 5.** Density lattice plots of the average spectral levels in an 1/3 octave band.

Densities overlap in almost all the charts did not allow a clear distinction between the three operating conditions. Only in some cases it was possible to distinguish a clear separation between the densities, but not for all three operating conditions. This was the case of the 125 Hz frequency, in which the density of the ON-H condition was clearly separated from the other two. Therefore, from the analysis of Figure 5 it was not possible to establish which frequencies allowed to make this distinction.

However, from the analysis of Figures 4 and 5, it was possible to understand that not all the frequencies available were necessary for the construction of our model; some of these could be left out. To confirm this, we can conduct a correlation analysis. For this purpose, the Pearson correlation index was calculated that expressed a possible linearity relationship between the variables. It is given by the sum of the products of the standardized scores of the two variables, divided by the number of subjects (or observations). A correlation matrix was calculated, but since there were thirty-one variables, its representation was difficult, at least in a tabular form. This could be helped by using a graphical display of a correlation matrix [29,30].

In Figure 6, the blue boxes indicate a strong correlation, while the white ones indicate a lack of correlation, according to the heat map, shown to the right of the figure. The positive correlations are displayed in blue and the negative correlations in red. Color intensity and the size of the square were proportional to the correlation coefficients. Furthermore, highly correlated variables were determined. A search through the correlation matrix was performed and the columns to be removed were returned, to reduce the correlations in pairs. In Table 2 the highly correlated variables, with the target, are shown.

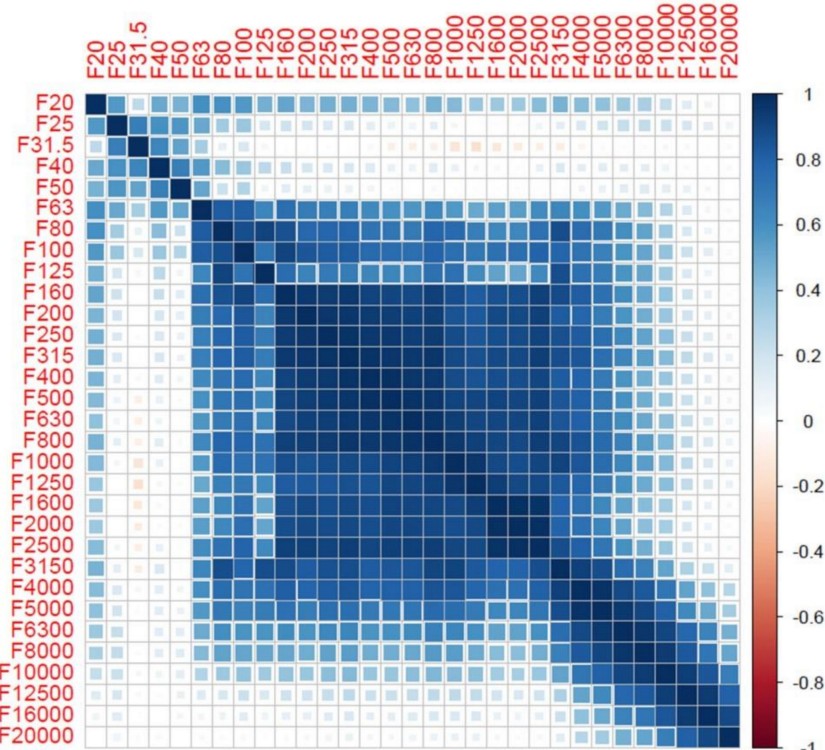

**Figure 6.** Graphical display of a correlation matrix.

**Table 2.** Highly correlated variables with the target (Hz).

| Frequency | Frequency | Frequency | Frequency | Frequency | Frequency | Frequency |
|-----------|-----------|-----------|-----------|-----------|-----------|-----------|
| 800       | 315       | 4000      | 160       | 3150      | 250       | 1000      |
| 200       | 500       | 630       | 400       | 2500      | 5000      | 100       |
| 1250      | 80        | 2000      | 6300      | 63        | 125       | 8000      |
| 10,000    | 20        | 12,500    | 25        | 40        | 20,000    | 50        |

To evaluate the importance of the variables, in order to rank features by importance, a Learning Vector Quantization (LVQ) model was set. The relative importance provided a score that indicated how useful or valuable each feature was, in the construction of the model.

The Learning Vector Quantization algorithm is a competitive network which uses supervised learning. It is a process of classifying patterns, where each output unit represents a class. The network is given a set of training patterns with a known classification, along with an initial distribution of the output class. After the training phase, this algorithm classifies an input vector by assigning it to the same class as that of the output unit. In the model setting, we divided our training dataset, randomly, into ten parts and then used each of the ten parts as a testing dataset, for the model, trained on the other nine. We took the average of the ten error terms, thus obtained. Then in three repeats of a ten-fold CV, the average of three error terms obtained by performing a ten-fold CV, five times, was performed. Values of the model tuning parameters were selected.

After training, the model variable importance was calculated. Sensitivity and specificity were computed for each cutoff and the ROC curve was computed. The trapezoidal rule was used to compute the area under the ROC curve. This area was used as a measure of variable importance. The variables were sorted by maximum importance across the classes—only the twenty most important variables were returned (out of thirty-one), as shown in Table 3.

**Table 3.** Variable importance.

| Frequency (Hz) | OFF | ON.H | ON.L |
|:---:|:---:|:---:|:---:|
| 2500 | 1.0000 | 1.0000 | 0.9996 |
| 80 | 0.9945 | 1.0000 | 1.0000 |
| 125 | 1.0000 | 1.0000 | 1.0000 |
| 3150 | 0.9895 | 0.9980 | 0.9980 |
| 160 | 0.9974 | 0.9974 | 0.9839 |
| 1600 | 0.9945 | 0.9944 | 0.9945 |
| 200 | 0.9900 | 0.9900 | 0.9885 |
| 2000 | 0.9889 | 0.9889 | 0.9889 |
| 315 | 0.9882 | 0.9882 | 0.9566 |
| 250 | 0.9856 | 0.9856 | 0.9629 |
| 800 | 0.9828 | 0.9828 | 0.9489 |
| 500 | 0.9819 | 0.9819 | 0.9533 |
| 400 | 0.9750 | 0.9750 | 0.9487 |
| 1000 | 0.9631 | 0.9733 | 0.9733 |
| 630 | 0.9721 | 0.9721 | 0.9481 |
| 100 | 0.9552 | 0.9552 | 0.9304 |
| 63 | 0.9531 | 0.9531 | 0.8568 |
| 4000 | 0.9524 | 0.9524 | 0.9322 |
| 1250 | 0.9449 | 0.9449 | 0.9174 |
| 5000 | 0.9323 | 0.9323 | 0.8902 |

Figure 7 shows the relative importance of all thirty-one variables, for each operating condition. The variable that recorded the highest values in importance was frequency, at 125 Hz. This frequency had already been identified, previously, through visual analysis, using the density latex plot, frequency was able to isolate the operating condition ON-H from the other two: The analysis confirmed this hypothesis. Furthermore, if we analyze the first twenty variables returned (importance > 0.9), we can see that they fall within the 50–5000 Hz frequency range. This range had already been previously identified (analyzing the frequency spectra of Figure 4) as being able to provide the information necessary to identify the operating conditions of the HVAC system.

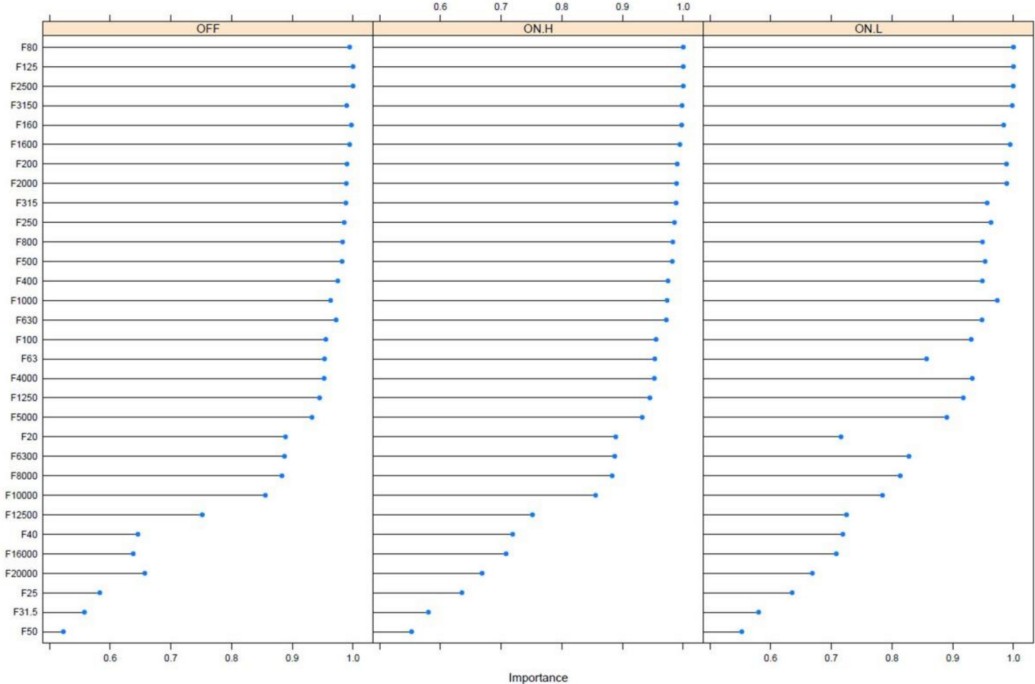

**Figure 7.** Relative importance of the variables for each operating condition.

The analysis of Figure 7 shows the possibility of selecting only a few variables to identify the operating conditions of the HVAC system.

To reduce the number of variables, a feature selection analysis was conducted. The purpose of the selection of features was to find features that allowed for a better discrimination of the different objects and which are insensitive to problems of translation, rotation, or scale. In other words, a reduction of the predictors was applied, which reduced the complexity of the information to be processed and made the system more efficient. An extracted parameter must be obtained in a simple manner and, at the same time, must have a high discriminating power. A recursive feature elimination (RFE) algorithm was used. This algorithm implemented the backwards selection of predictors, based on the predictor importance ranking. The predictors were ranked and the less important ones were sequentially eliminated, prior to modeling. The goal was to find a subset of predictors that could be used to produce an accurate model. For the outer resampling method, Cross-Validation (ten-fold) was used. Table 4 shows the resampling performance over a subset size.

**Table 4.** Resampling performance over a subset size.

| Variables | Accuracy | Kappa | Accuracy SD |
|:---:|:---:|:---:|:---:|
| 1 | 0.8130 | 0.7194 | 0.076329 |
| 2 | 0.9389 | 0.9083 | 0.019618 |
| 3 | 0.9963 | 0.9944 | 0.007808 |
| 4 | 0.9981 | 0.9972 | 0.005856 |
| 5 | 0.9981 | 0.9972 | 0.005856 |
| 6 | 0.9981 | 0.9972 | 0.005856 |
| 7 | 0.9981 | 0.9972 | 0.005856 |
| 8 | 0.9981 | 0.9972 | 0.005856 |
| 9 | 1.0000 | 1.0000 | 0.000000 |
| 10 | 0.9981 | 0.9972 | 0.005856 |
| 11 | 0.9981 | 0.9972 | 0.005856 |
| 12 | 1.0000 | 1.0000 | 0.000000 |
| 13 | 0.9981 | 0.9972 | 0.005856 |
| 14 | 1.0000 | 1.0000 | 0.000000 |
| 15 | 0.9981 | 0.9972 | 0.005856 |
| 16 | 0.9981 | 0.9972 | 0.005856 |
| 17 | 0.9981 | 0.9972 | 0.005856 |
| 18 | 0.9981 | 0.9972 | 0.005856 |
| 19 | 0.9981 | 0.9972 | 0.005856 |
| 20 | 0.9981 | 0.9972 | 0.005856 |
| 21 | 0.9981 | 0.9972 | 0.005856 |
| 22 | 0.9981 | 0.9972 | 0.005856 |
| 23 | 0.9981 | 0.9972 | 0.005856 |
| 24 | 0.9981 | 0.9972 | 0.005856 |
| 25 | 0.9981 | 0.9972 | 0.005856 |
| 26 | 0.9981 | 0.9972 | 0.005856 |
| 27 | 1.0000 | 1.0000 | 0.000000 |
| 28 | 1.0000 | 1.0000 | 0.000000 |
| 29 | 0.9981 | 0.9972 | 0.005856 |
| 30 | 1.0000 | 1.0000 | 0.000000 |
| 31 | 0.9981 | 0.9972 | 0.005856 |

The Accuracy column gives the average accuracy of the thirty-one variables and the column labeled as Accuracy SD gives the standard deviation of the thirty-one variable accuracies. The Kappa statistic is a measure of concordance, for the categorical data, which measures agreement relative to what would be expected by chance. Values of 1 indicate perfect agreement, while a value of zero would indicate a lack of agreement. Kappa is an excellent performance measure, when the classes are highly unbalanced.

Metrics are required to determine whether the adopted method works. Model performance was estimated using resampling. Resampling methods are essential for testing and evaluating statistical models. This technique repeatedly extracts only a portion of the data, from the sample, to get more information about the model, instead of repeatedly sampling the entire data set. This makes it possible to evaluate the variability and stability of the model. Cross-validation estimates the different adaptation measures of the model, taking a subset of the original training observations from the model training process, and then applying that statistical learning method to the unused sets. Figure 8 shows the resampling results for the candidate subset sizes, evaluated during the recursive feature elimination (RFE) process [19].

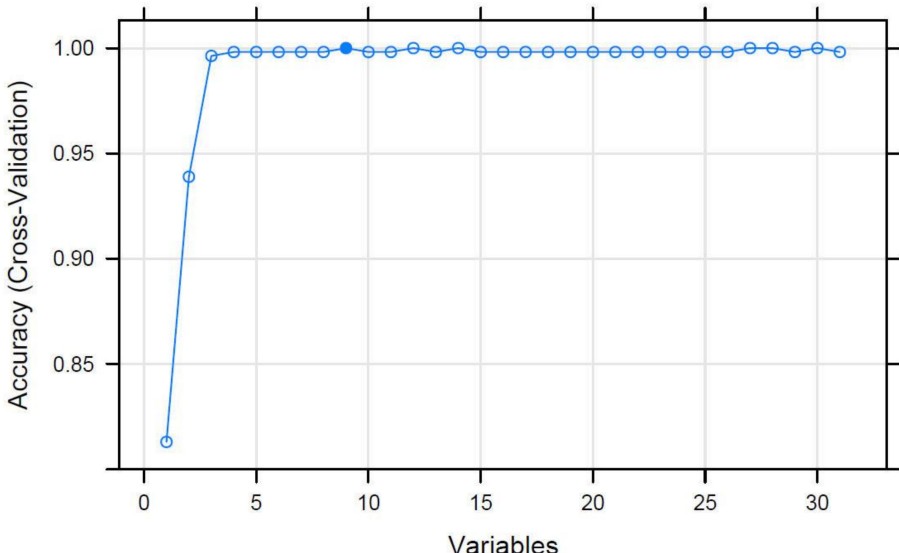

**Figure 8.** Resampling results for the candidate subset sizes.

The accuracy obtained ensured that the adopted model works. The variables selected by the model are listed in Table 5.

**Table 5.** Variables selected (Frequency, Hz).

| 125 | 80 | 2500 | 1600 | 2000 | 160 | 3150 | 200 | 100 |
|-----|----|------|------|------|-----|------|-----|-----|

The variables selected by the RFE method represents the input variables for a recursive partitioning model, implemented in order to detect the operating conditions of the HVAC system. This model creates a decision tree to classify categorical data, by splitting it into sub-systems, based on several dichotomous independent variables.

Analyzing Figure 9, it was possible to notice that, as the root node, the frequency of 125 Hz was identified, which represented the frequency with the greatest relative importance (Table 3). A Leq >43.6 dB, in this frequency, already allowed us to identify an operating condition (ON-H). For values <43.6 we went to another node that identified with the frequency of 2500 Hz. A Leq < 35.1 dB, in this frequency, allowed us to identify another operating condition (OFF). For values >35.1, we went to another node that identified with the frequency of 1600 Hz. A Leq >35.9 dB, in this frequency, allowed us to identify the third operating condition (ON-L). For Leq <35.9 dB, two already identified operating conditions were recognized as indicating a system error.

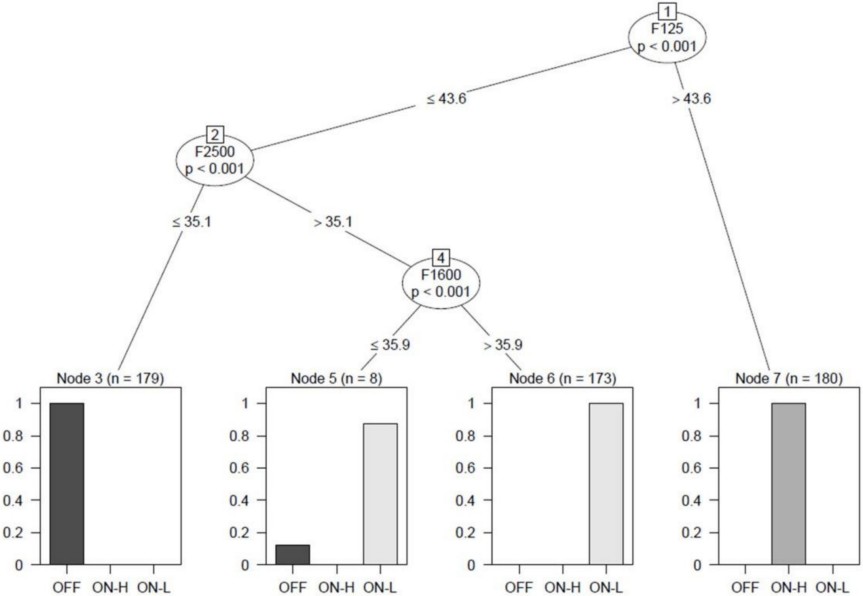

**Figure 9.** Conditional Inference Tree for the HVAC data. Each node tests a condition (test) on a particular property of the sound source (Frequency) and has two or more branches downwards. The process consists of a sequence of tests. Always start from the root node, the node located higher up in the structure, and proceed downwards. Depending on the values measured at each node, the flow takes one direction, or another, and proceeds progressively downwards.

Table 6 shows a confusion matrix of the model. A confusion matrix is a table that is used to describe the performance of a classification model on a set of data, for which the true values are known. The rows correspond to the predicted class and the columns correspond to the true class. The diagonal cells correspond to the observations that were correctly classified. The off-diagonal cells correspond to the incorrectly classified observations. The number of observations has been shown in each cell.

**Table 6.** Confusion matrix.

| Predictions | OFF | ON-H | ON-L |
|---|---|---|---|
| OFF | 179 | 0 | 0 |
| ON-H | 0 | 180 | 0 |
| ON-L | 1 | 0 | 180 |

The confusion matrix returned excellent results—a single error. The accuracy of the model was very high (Accuracy = 0.9981), confirming that the recursive partitioning was able to identify the operating conditions of an HVAC system. Table 7 lists several associated statistics of the confusion matrix.

**Table 7.** Associated statistics of the confusion matrix.

| Class | OFF | ON-H | ON-L |
|---|---|---|---|
| Sensitivity | 0.9944 | 1.0000 | 1.0000 |
| Specificity | 1.0000 | 1.0000 | 0.9972 |
| Pos Pred Value | 1.0000 | 1.0000 | 0.9945 |
| Neg Pred Value | 0.9972 | 1.0000 | 1.0000 |
| Prevalence | 0.3333 | 0.3333 | 0.3333 |
| Detection Rate | 0.3315 | 0.3333 | 0.3333 |
| Detection Prevalence | 0.3315 | 0.3333 | 0.3352 |
| Balanced Accuracy | 0.9972 | 1.0000 | 0.9986 |

In Table 7, the terms assume the following meanings. Sensitivity measures the proportion of actual positives that are correctly identified as such. It is calculated as the number of correct positive predictions divided by the total number of positives. The best sensitivity is 1.0, whereas, the worst is 0.0. Specificity measures the proportion of actual negatives that are correctly identified as such. It is calculated as the number of correct negative predictions divided by the total number of negatives. The best specificity is 1.0, whereas the worst is 0.0. Pos Pred Value and Neg Pred Value return the number of positive and negative forecasts obtained. The prevalence measures how many times the positive condition occurs in our sample. The Detention rate is a parameter that will vary according to the dataset. It is calculated as the number of correct positive predictions, divided by the total number of the sample. Detention prevalence is calculated as the number of total positive predictions, divided by the total number of the sample. The Balanced Accuracy represents the arithmetic mean between the sensitivity and specificity.

## 4. Conclusions

In this study, measurements were taken of the noise emitted by the HVAC system, in an open-plan office. The average spectral levels in an 1/3 octave band were compared through a correlation analysis, to identify redundant data. A model was then adapted to evaluate the importance of variables, in order to classify the characteristics, by importance. To reduce the number of predictor variables, a selection analysis of the characteristics was carried out. Thus, a subset of predictors was extracted to be used, to produce an accurate prediction model. Finally, a model based on recursive partitioning for the detection of the operating conditions of an HVAC system was developed and applied, to provide insights into the development and application of this technique, in these contexts. In particular, the high accuracy of the model (Accuracy = 0.9981) suggests the adoption of this tool for several applications. This methodology can be used for the automatic recognition of the operating conditions of an HVAC system, in order to monitor the energy consumption of the plant. It can also be used to recognize the sources responsible for unwanted noise emissions in workplaces.

**Author Contributions:** All the authors contributed to the original idea and design of the study, to the analysis, to the drafting of the manuscript, reading and approving the final version.

**Funding:** This research received no external funding.

**Conflicts of Interest:** The authors declare no conflicts of interest.

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
