# Peer review of "Heating, Ventilation, and Air Conditioning (HVAC) Noise Detection in Open-Plan Offices Using Recursive Partitioning"

_buildings, doi:10.3390/buildings8120169_

Round 1
Reviewer 1 Report
The paper contains interesting information about HVAC noise detection in open-plan offices and indicates the effectiveness of recursive partitioning, which justifies publication.
My comments:
1. You cannot start the paper title with an acronym. Explain also the acronym in the abstract.
2. You do not demonstrate adequate literature in the field. Is there any research relevant to HVAC noise?
3. The methodology is well described and the results are presented clearly and analyzed appropriately. But, you do not discuss them adequately in light of the relevant literature.
4. You do not clarify how the results could be used by professionals or in practice.
Author Response
1. You cannot start the paper title with an acronym. Explain also the acronym in the abstract. (We have modified the title as suggested, thanks for the useful suggestion)
2. You do not demonstrate adequate literature in the field. Is there any research relevant to HVAC noise? (I added adequate literature, thanks for the useful suggestion)
3. The methodology is well described and the results are presented clearly and analyzed appropriately. But, you do not discuss them adequately in light of the relevant literature. (I added an appropriate discussion with references to the literature)
4. You do not clarify how the results could be used by professionals or in practice. (I have added example of practice use of the methodology)

Reviewer 2 Report
In the paper the results of measurements of the noise emitted by the HVAC system in an open-plan office are presented.
The methods used by the authors are appropriate but there are some points that should be addressed before this paper is accepted for the publication.
Section “Methodology”
Add more information about measuring room, e.g. room height, sound absorption in the room.
In Figure 2, indicate the location selected to measurement.
Section “Results”
Table 1 Change 48,6 to 48.6
There is no discussion of Figure 9 in the text. Add the appropriate paragraph.
Figures 7, 8 and 9 are unreadable. Change the axes descriptions in particular.
Section “Conclusions”
This section only describes what has been done in the paper. The section should contain the main conclusions from the work. Add the appropriate paragraph.
Author Response
Section “Methodology”
Add more information about measuring room, e.g. room height, sound absorption in the room. (I have added room dimensions, I have also included a brief description of the furnishing accessories.)
In Figure 2, indicate the location selected to measurement. (I have added measurement point location)
Section “Results”
Table 1 Change 48,6 to 48.6 (Thank you, I modified)
There is no discussion of Figure 9 in the text. Add the appropriate paragraph. (I have added appropriate paragraph and some text in the caption)
Figures 7, 8 and 9 are unreadable. Change the axes descriptions in particular. (I have elaborated once again the figures)
Section “Conclusions”
This section only describes what has been done in the paper. The section should contain the main conclusions from the work. Add the appropriate paragraph. (I have added example of practice use of the methodology)
